# Modification of Conventional Sugar Juice Evaporation Process for Increasing Energy Efficiency and Decreasing Sucrose Inversion Loss

**Somchart Chantasiriwan**

Department of Mechanical Engineering, Thammasat University, Pathum Thani 12121, Thailand; somchart@engr.tu.ac.th

**Abstract:** The evaporation process, boiler, and turbine are the main components of the cogeneration system of the sugar factory. In the conventional process, the evaporator requires extracted steam from the turbine, and bled vapor from the evaporator is supplied to the juice heater and the pan stage. The evaporation process may be modified by using extracted steam for the heating duty in the pan stage. This paper is aimed at the investigation of the effects of this process modification. Mathematical models of the conventional and modified processes were developed for this purpose. It was found that, under the conditions that the total evaporator area is 13,000 m$^2$, and the inlet juice flow rate is 125 kg/s, the optimum modified evaporation process requires extracted steam at a pressure of 157.0 kPa. Under the condition that the fuel consumption rate is 21 kg/s, the cogeneration system that uses the optimum modified evaporation process yields 2.3% more power output than the cogeneration system that uses a non-optimum conventional cogeneration process. Furthermore, sugar inversion loss of the optimum modified process is found to be 63% lower than that of the non-optimum conventional process.

**Keywords:** heat exchanger; mathematical model; energy efficiency; inversion loss; process design; mass transfer

## 1. Introduction

The evaporation process, boiler, and steam turbine are the main components of the cogeneration system in the sugar industry. Diluted sugar juice becomes raw sugar and molasses in the evaporation process after a specified amount of water is removed by evaporation. Thermal energy required for water evaporation is provided by steam condensation. The boiler generates high-pressure steam that is supplied to the steam turbine for power generation. Older cogeneration systems use the back-pressure turbine, in which steam is exhausted at a lower pressure, whereas modern cogeneration systems use the extraction–condensing steam turbine, in which some steam is extracted at lower pressure, and the remaining steam is sent to the condenser. Kamate and Gangavati [1] have shown that a cogeneration system using the extraction–condensing steam turbine is more energy efficient than a cogeneration system using the back-pressure steam turbine.

The multiple-effect evaporator is used in the evaporation process. The evaporator is designed to increase the juice concentration from approximately 15% to 70%. The removal of the remaining water content in sugar occurs in the pan stage. The multiple-effect evaporator requires a supply of saturated steam extracted from an extraction–condensing steam turbine at a specified pressure. An adverse consequence of the exposure of sugar juice to high-temperature steam and vapor in the multiple-effect evaporator is sucrose inversion loss, which converts sucrose to glucose and fructose. In order to increase the profitability of raw sugar manufacturing, the amount of required steam

and sugar inversion loss should be minimized without compromising the capacity of the process. There have been several suggestions to improve the energy efficiency of the evaporation process. Urbaniec et al. [2] have suggested that heat recovery can be improved by retrofitting the evaporation process. Ensinas et al. [3] have used a thermo-economic procedure to reduce steam consumption by the evaporation process. An analysis by Higa et al. [4] shows that increasing the number of effects can decrease steam consumption. Bapat et al. [5] have shown that steam consumption can be reduced by using heat recovery devices. Sharan and Bandyopadhyay [6] have shown that steam consumption by the entire plant can be minimized by integrating the evaporator with the background process. Mechanical vapor compression [7] and thermal vapor compression [8] have been suggested as methods to increase the energy efficiency of the multiple-effect evaporator. The energy efficiency of the multiple-effect evaporator can also be increased by the optimum distribution of heating surface areas [9–12]. Recently, Chantasiriwan has shown that the energy efficiency of the cogeneration system, in which the evaporation process is a component, can be increased by replacing the forward-feed evaporator with the backward-feed evaporator [13]. Investigations of sucrose inversion loss in sugar juice evaporation process have yielded conclusions that increased time between cleanings of the evaporator results in more inversion loss [14]; inversion loss may be reduced by replacing Robert evaporators with falling-film evaporators [15]; and using smaller diameter and longer tubes decrease inversion loss due to shorter residence time of juice in the evaporator [16]. In addition, Rein [17] has suggested that decreasing the temperature profile across the effects of the evaporator can also reduce inversion loss.

Energy efficiency of the evaporation process can be improved not only by reducing the steam consumption of a given pressure, which is the subject of most of the previous investigations, but also by decreasing extracted steam pressure. There is a lower limit of extracted steam pressure because the thermal energy input required for an evaporation process is approximately equal to the product of the total heating surface area of the evaporator and the difference between the steam temperature at the evaporator inlet and the vapor temperature at the evaporator outlet. The lower limit can be decreased by increasing the total heating surface area. Furthermore, it is interesting to note that vapor is usually bled from the first effect of the multiple-effect evaporator in order to be used for heating duty in the pan stage. This requirement imposes an additional constraint on the lower limit of the extracted steam pressure. It is possible to remove this constraint by using extracted steam instead of bled vapor for this purpose. A consequence of this constraint removal is further reduction of extracted steam pressure. Reduced extracted steam pressure results in not only higher energy efficiency but also lower sucrose inversion loss due to decreased temperature profile across the effects of the evaporator [17].

In this paper, the performance of the conventional process, in which vapor bled from the multiple-effect evaporator is used for the pan stage, is compared with that of a modified sugar juice evaporation process, which uses extracted steam instead of bled vapor for heating duty in the pan stage. Mathematical models of the conventional and modified processes are presented in Sections 2 and 3. Both processes operate in cogeneration systems described in Section 4. Section 5 shows that, under the same conditions, differences in energy efficiency and sucrose inversion loss can be attributed to the process modification.

## 2. Conventional Evaporation Process

The conventional sugar juice evaporation process is shown in Figure 1. The components of the process are 4 effects of the evaporator (E1, E2, E3, and E4), 2 heat exchangers of the juice heater (H1 and H2), the flash tanks (FC, F1, F2, and F3), and the pan stage (P). Sugar juice at the ambient temperature ($T_{h,2}$) is heated in H2 and H1 to the saturation temperature ($T_{h,0}$), which is 103 °C. This temperature corresponds to a pressure slightly larger than the atmospheric pressure ($p_{atm}$). Juice pressure is decreased to $p_{atm}$ in FC before entering E1. Sugar juice and saturated steam or vapor flow from E1 to E4. The steam turbine (not shown in Figure 1) supplies extracted steam at pressure $p_0$ to E1. Vapor from E1 is sent to P, H1, and E2. Vapor from E2 is sent to H2 and E3. Vapor from E3 is sent to E4. Vapor from

E4 is sent to the condenser (not shown in Figure 1). In effect *i*, water evaporation at pressure $p_{i+1}$ is caused by vapor condensation at pressure $p_i$. Concentrated sugar juice from E4 is sent to P.

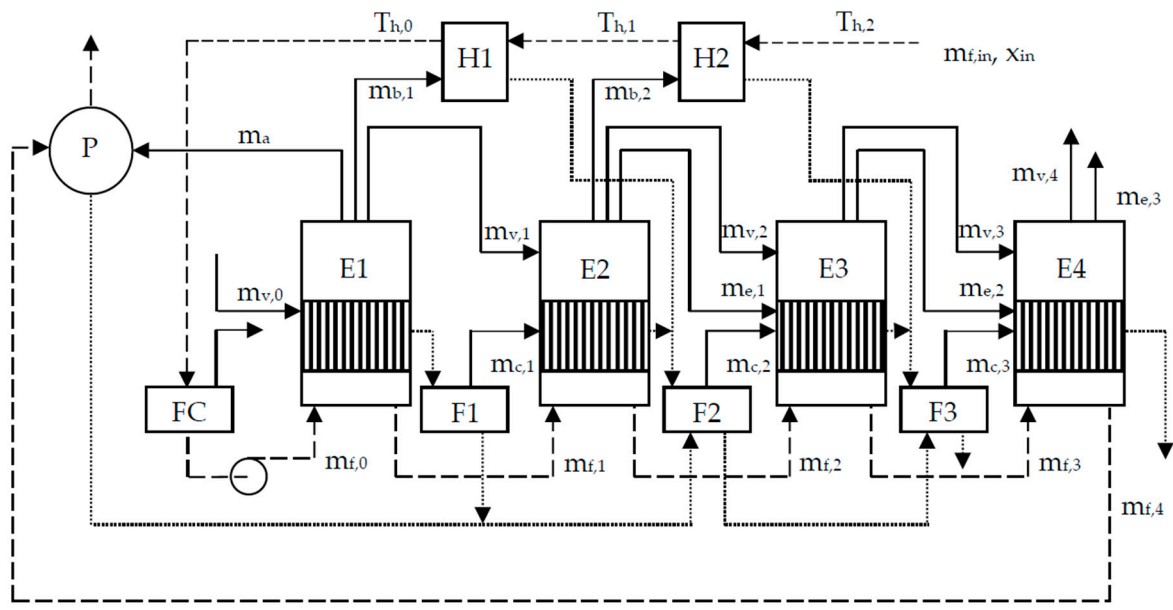

**Figure 1.** Conventional evaporation process.

In order to improve the energy efficiency of the process, condensates from E1, E2, and E3 are sent, respectively, to F1, F2, and F3. F2 also receives condensates from P, F1, and H1, and F3 also receives condensates from F2 and H2. Flash tanks (F1, F2, and F3) produce vapor and condensate at a lower pressure from condensate at a higher pressure.

The model of the conventional evaporation process in Figure 1 is similar to the model presented by Chantasiriwan [13]. The difference between the two models is the treatment of condensate from E1. In the model presented by Chantasiriwan [13], the condensate is sent to the boiler. In the model shown in Figure 1, the condensate is sent to F1. It can be shown that this treatment increases the overall energy efficiency of the process.

Due to the similarity between this model and the model presented by Chantasiriwan [13], only different equations are shown for the sake of concise presentation. The different treatment of condensate in this paper gives rise to the following energy equations:

$$(1-\varepsilon)(m_{v,0}-m_{x,0})h_{vl}(p_0) + m_{f,0}\left(h_{f,1}^{(in)} - h_{f,1}^{(out)}\right) = \left(m_a + m_{v,1} + m_{b,1}\right)\left[h_v(p_1) - h_{f,1}^{(out)}\right], \tag{1}$$

$$(1-\varepsilon)(m_{v,1}+m_{c,1})h_{vl}(p_1) + m_{f,1}\left(h_{f,2}^{(in)} - h_{f,2}^{(out)}\right) = \left(m_{v,2} + m_{b,2}\right)\left[h_v(p_2) - h_{f,2}^{(out)}\right], \tag{2}$$

$$(1-\varepsilon)(m_{v,2}+m_{e,1}+m_{c,2})h_{vl}(p_2) + m_{f,2}\left(h_{f,3}^{(in)} - h_{f,3}^{(out)}\right) = m_{v,3}\left[h_v(p_3) - h_{f,3}^{(out)}\right], \tag{3}$$

$$(1-\varepsilon)(m_{v,3}+m_{e,2}+m_{c,3})h_{vl}(p_3) + m_{f,3}\left(h_{f,4}^{(in)} - h_{f,4}^{(out)}\right) = m_{v,4}\left[h_v(p_4) - h_{f,4}^{(out)}\right], \tag{4}$$

$$m_{c,1} = m_{v,0}f(p_0,p_1), \tag{5}$$

$$m_{c,2} = \left(m_{v,0} + m_{v,1} + m_{b,1} + m_a\right)f(p_1,p_2), \tag{6}$$

$$m_{c,3} = \left(m_{v,0} + m_a + m_{v,1} + m_{b,1} + m_{v,2} + m_{b,2} + m_{e,1}\right)f(p_2,p_3). \tag{7}$$

Expressions for the other parameters are the same as those in the model presented by Chantasiriwan [13], and an interested reader is asked to consult that reference. The heat

transfer equations in this model are also slightly different from those in the model presented by Chantasiriwan [13]. They are shown as follows.

$$U_1(A_1 - A_{x,0})\left[T_{sat}(p_0) - T_{f,1}^{(out)}\right] = (1-\varepsilon)(m_{v,0} - m_{x,0})h_{vl}(p_0). \tag{8}$$

$$U_2 A_2\left[T_{sat}(p_1) - T_{f,2}^{(out)}\right] = (1-\varepsilon)(m_{v,1} + m_{c,1})h_{vl}(p_1), \tag{9}$$

$$U_3 A_3\left[T_{sat}(p_2) - T_{f,3}^{(out)}\right] = (1-\varepsilon)(m_{v,2} + m_{e,1} + m_{c,2})h_{vl}(p_2), \tag{10}$$

$$U_4 A_4\left[T_{sat}(p_3) - T_{f,4}^{(out)}\right] = (1-\varepsilon)(m_{v,3} + m_{e,2} + m_{c,3})h_{vl}(p_3). \tag{11}$$

## 3. Modified Evaporation Process

The conventional evaporation process uses bled vapor from the first effect of the evaporator for the pan stage. A consequence of this requirement is that the extracted steam pressure ($p_0$) must not be lower than the minimum value that corresponds to a specified juice mass flow rate. It is possible to remove this constraint by using extracted steam instead of bled vapor for the pan stage in the modified evaporation process.

The modified evaporation process is depicted in Figure 2. It can be seen that extracted steam at pressure $p_a$ is supplied to the pan stage. The model of this process is the same as that of the conventional process with $m_a$ deleted from Equation (1). The mass flow rate of extracted steam required by the pan stage is

$$m_a = \frac{2m_{f,4}(1 - x_4/91)h_{vl}(p_4)}{h_{vl}(p_a)}. \tag{12}$$

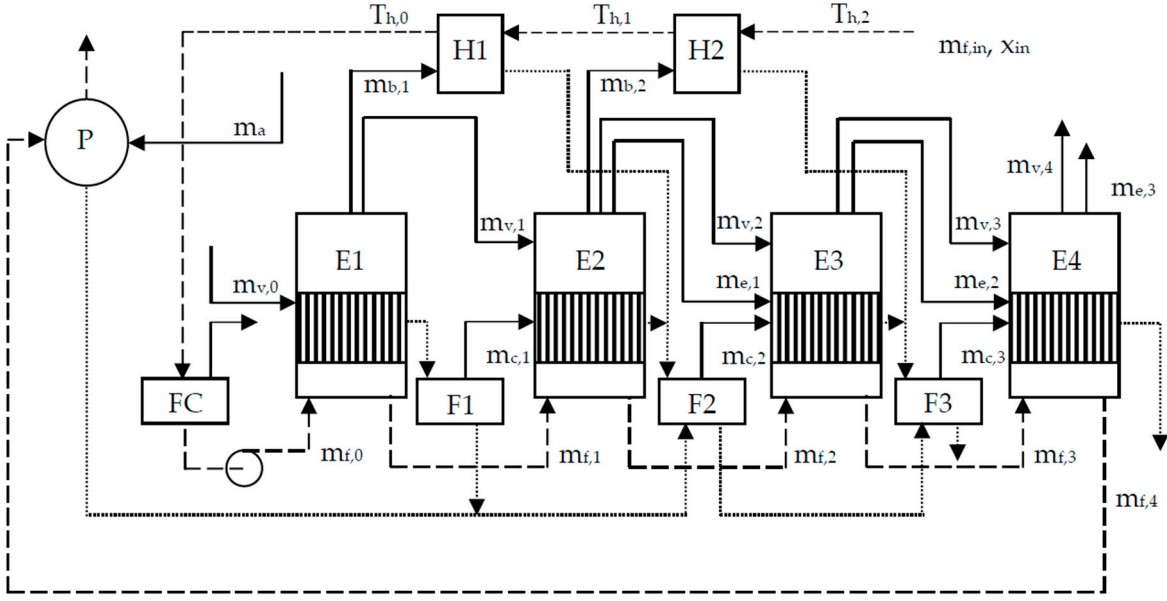

**Figure 2.** Modified evaporation process.

It is interesting to note that, under the same operating conditions, $m_{f,4}$, $x_4$, and $p_4$ of the modified and conventional evaporation processes are identical. Therefore, the values of $m_a$ of both processes are the same if $p_a = 150$ kPa.

## 4. Performance Parameters

This paper is intended to demonstrate that the modified evaporation process requires extracted steam at a lower pressure than the conventional evaporation process, which leads to the enhanced performance of the modified evaporation process compared with the conventional process. Comparison between both processes is based on two performance parameters, which are turbine power output of the cogeneration system and sucrose inversion loss.

### 4.1. Turbine Power Output

Steam economy is the performance parameter that may be used to evaluate the energy efficiency of an evaporation process. It is equal to the ratio of the mass flow rate of evaporated water to the mass flow rate of extracted steam. Therefore,

$$SE = \frac{2m_{f,in}(1 - x_4/91)}{m_{v,0}} \tag{13}$$

for the conventional evaporation process, and

$$SE = \frac{2m_{f,in}(1 - x_4/91)}{m_{v,0} + m_a} \tag{14}$$

for the modified evaporation process.

Steam economy is an appropriate parameter for comparing different conventional evaporation processes because the extracted steam pressure in the first effect of the multiple-effect evaporator is fixed. The process having larger steam economy is considered to be more energy efficient. However, steam economy should not be used to compare the conventional and modified evaporation processes because extracted steam pressures in both processes may be different. To identify a more suitable performance parameter, it is necessary to consider the cogeneration system.

The cogeneration systems for the conventional and modified evaporation processes are depicted in Figure 3. In each system, the mass flow rate, pressure, and temperature of steam generated by the boiler (B) are, respectively, $m_s$, $p_s$, $T_s$. Steam is extracted at the pressure of $p_0$ in the conventional evaporation process. The mass flow rate of extracted steam is $m_{v,0}$. The extracted steam is used for evaporation in the first effect of the evaporator. The remaining steam is condensed at the pressure of $p_c$. The mass flow rate of condensed steam ($m_c$) is, therefore, $m_s - m_{v,0}$. The modified evaporation process requires not only extracted steam at the pressure of $p_0$ for evaporation in the first effect of the evaporator but also extracted steam at the pressure of $p_a$ for evaporation in the pan stage. The corresponding mass flow rates of extracted steam are $m_{v,0}$ and $m_a$. The remaining steam is condensed at the pressure of $p_c$. The mass flow rate of condensed steam ($m_c$) is, therefore, $m_s - m_{v,0} - m_a$.

Inspection of Figure 3 reveals that the inputs of both systems are sugar juice and bagasse, and the outputs are turbine power, sugar, and molasses. Both systems are assumed to have the same juice processing capacity. This means that $m_{f,in}$, $x_{in}$, and $x_4$ are the same in both the conventional evaporation process and the modified evaporation process. Moreover, both systems are assumed to consume the same amount of fuel ($m_{fuel}$) in their boilers. Based on these assumptions, the only difference between both systems is turbine power output, which is expressed as

$$P = m_{v,0}(h_s - h_0) + m_a(h_s - h_a) + m_c(h_s - h_c), \tag{15}$$

$$h_0 = h_s - \eta_t(h_s - h_{0s}), \tag{16}$$

$$h_a = h_s - \eta_t(h_s - h_{as}), \tag{17}$$

$$h_c = h_s - \eta_t(h_s - h_{cs}), \tag{18}$$

where $\eta_t$ is isentropic efficiency of the steam turbine, $h_s$ is specific enthalpy at pressure $p_s$, and temperature $T_s$, $h_{0s}$, $h_{as}$, and $h_{cs}$ are specific enthalpies at, respectively, pressures $p_0$, $p_a$, and $p_c$, and the same entropy as the inlet steam. It should be noted that $m_a$ is zero in the cogeneration system for the conventional evaporation process.

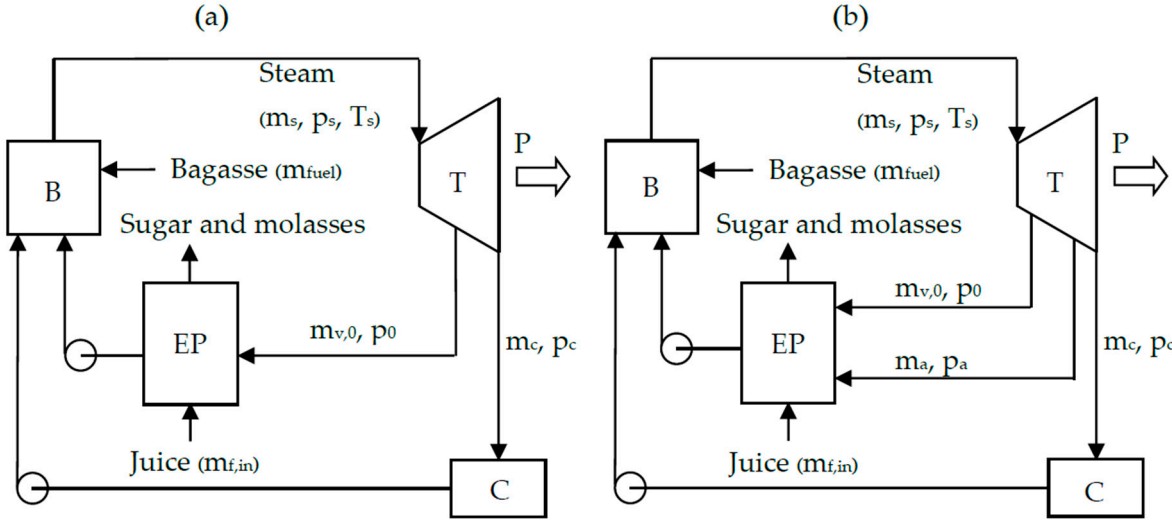

**Figure 3.** Cogeneration systems for (**a**) the conventional evaporation process and (**b**) the modified evaporation process.

*4.2. Sucrose Inversion Loss*

Sucrose inversion is the chemical reaction that transforms sucrose into glucose and fructose, which do not crystallize and cannot be recovered as sugar. Main factors that influence sucrose inversion in the multiple-effect evaporator are temperature, time, juice acidity, and juice concentration. Sucrose inversion loss may be estimated by using the Vukov model [18]. According to this model, the mass fraction of lost sucrose is expressed as

$$I = 1 - e^{-kt}, \tag{19}$$

where $t$ is retention time (in minutes) of sugar juice in an evaporator vessel. The reaction rate ($k$) is determined from

$$\log k = 16.91 - \log\left[\frac{\rho(100 - x)}{100}\right] - \frac{5670}{T_f} - pH. \tag{20}$$

Equation (20) is applicable when the juice temperature is 25 °C. At a different temperature, the corrected *pH* value is

$$pH = pH_{25} + \left(T_f - 25\right)\left(-0.0339 + 0.015 pH_{25} - 0.0017 pH_{25}^2\right). \tag{21}$$

For simulation purposes, $pH_{25}$ is assumed to be 6.0. The retention time ($t$) is proportional to the evaporator surface area ($A$), and inversely proportional to sugar juice mass flow rate ($m_f$). It may be approximated by assuming that sugar juice flows through $N$ tubes, of which diameter and length are $D$ and $L$, in an evaporator vessel at the speed of $V$. The expression of $V$ is

$$V = \frac{4m_f}{N\rho\pi D^2}. \tag{22}$$

Consequently,

$$t = \frac{N\rho\pi D^2 L}{240 m_f}. \tag{23}$$

If tube thickness is negligible, the heating surface of the evaporator vessel ($A$) is $N\pi DL$, and Equation (23) becomes

$$t = \frac{\rho DA}{240m_f}. \tag{24}$$

Typical tube diameter varies from 38 to 51 mm. It is assumed that $D$ is 45 mm in this paper.

## 5. Results and Discussion

The parameters of both evaporation processes are $x_{in}$ = 15%, $x_{out}$ = 70%, $p_4$ = 16 kPa, and $T_{h,2}$ = 30 °C. In each process, the total surface areas of the multiple-effect evaporator and the juice heater are, respectively, 13,000 and 2500 m². Multiple-effect evaporators in both systems are designed to process 125 kg/s (or 450 t/h) of juice. The optimum distribution of the total evaporator surface area that maximizes the steam economy at a specified extracted steam pressure ($p_0$) may be determined for each system.

The procedure for determining the optimum distribution of the evaporator surface area in the conventional evaporation process that maximizes the steam economy ($SE$) is shown in Figure 4. Figure 4a shows that, for the first-effect area ($A_1$) of 6000 m² and the second-effect area ($A_2$) of 1200 m², the optimum value of the third-effect area ($A_3$) that yields the required juice mass flow rate of 125 kg/s and the maximum steam economy ($SE$) is 1233 m². Figure 4b shows that, for the same value of $A_1$, the optimum value of $A_2$ that results in maximum $SE$ is 1251 m². Figure 4c shows that, as $A_1$ increases, $SE$ decreases, and first-effect pressure ($p_1$) increases. By requiring that $p_1$ is 150 kPa, the optimum value of $A_1$ is found to be 4518 m². The corresponding value of $SE$ is 2.508. Therefore, the mass flow rate of extracted steam for the evaporator ($m_{v,0}$) is 41.63 kg/s.

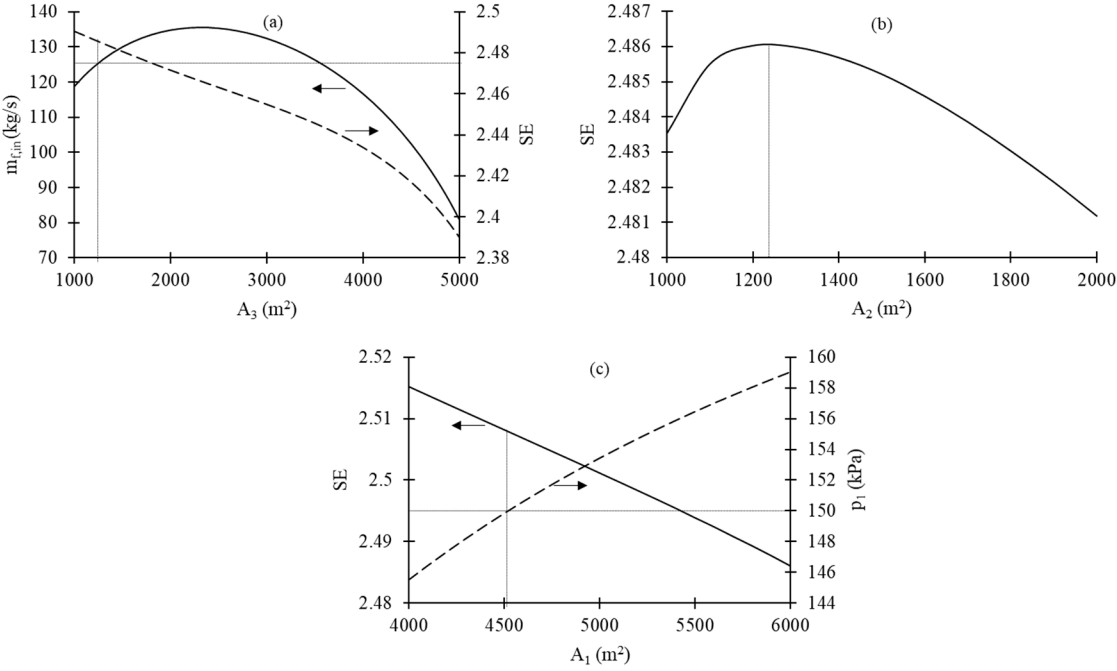

**Figure 4.** Procedure for determining the optimum distribution of evaporator area in the conventional evaporation process driven by extracted steam at a pressure ($p_0$) of 200 kPa: (**a**) finding third-effect area ($A_3$) corresponding to the inlet juice mass flow rate ($m_{f,in}$) of 125 kg/s and the maximum steam economy ($SE$) corresponding to first-effect area ($A_1$) = 6000 m², and second-effect area ($A_2$) = 1200 m²; (**b**) finding $A_2$ that maximizes $SE$ corresponding to $A_1$ = 6000 m²; and (**c**) finding $A_1$ corresponding to the first-effect pressure ($p_1$) of 150 kPa.

The procedure for determining the optimum distribution of evaporator surface area in the modified evaporation process that maximizes $SE$ is shown in Figure 5. Figure 5a shows that, for the first-effect area ($A_1$) of 4000 m$^2$ and the second-effect area ($A_2$) of 1100 m$^2$, the optimum value of the third-effect area ($A_3$) that yields the required juice mass flow rate of 125 kg/s and the maximum $SE$ is 1723 m$^2$. Figure 5b shows that, for the same value of $A_1$, the optimum value of $A_2$ that results in the maximum $SE$ is 1342 m$^2$. Figure 5c shows the optimum value of $A_1$ that results in the maximum $SE$ is 2074 m$^2$. The corresponding value of $SE$ is 2.345. Since the mass flow rate of juice leaving E4 ($m_{f,4}$) is 26.79 kg/s, and the mass flow rate of extracted steam for the pan stage ($m_a$) is 13.16 kg/s, the value of $m_{v,0}$ is found to be 31.53 kg/s.

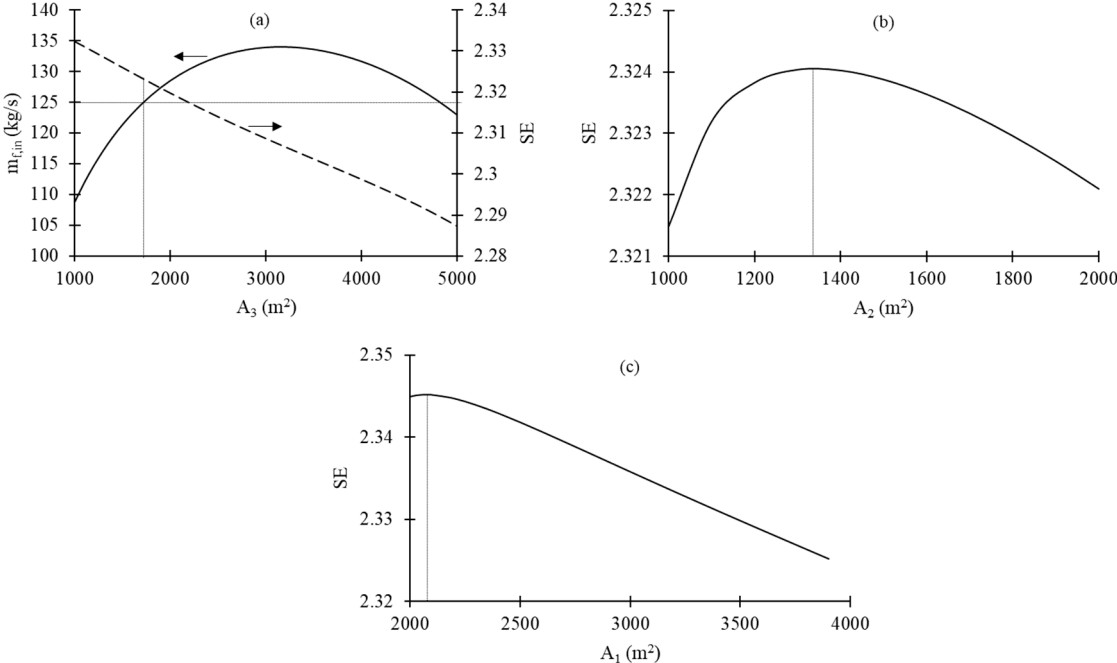

**Figure 5.** Procedure for determining the optimum distribution of evaporator area in the modified evaporation process driven by extracted steam at a pressure ($p_0$) of 200 kPa: (**a**) finding $A_3$ corresponding to the inlet juice mass flow rate ($m_{f,in}$) of 125 kg/s and the maximum steam economy ($SE$) corresponding to $A_1$ = 4000 m$^2$, and $A_2$ = 1100 m$^2$; (**b**) finding $A_2$ that maximizes $SE$ corresponding to $A_1$ = 4000 m$^2$; and (**c**) finding $A_1$ that maximizes $SE$.

The calculation of the turbine power output ($P$) of a cogeneration system requires information about the fuel, the boiler, and the steam turbine. It is assumed that the fuel consumption rate in the boiler of each system is 21 kg/s, the higher heating value of fuel is 9000 kJ/kg, the boiler efficiency is 70%, the pressure and temperature of superheated steam generated by the boiler are 4.5 MPa and 440 °C, and the turbine efficiency is 85%. Figure 6 shows variations of $m_{v,0}$ and $P$ with $p_0$ in cogeneration systems for the conventional and modified evaporation processes that have the optimum distributions of evaporator surface areas. It can be seen that, in each system, there exists the optimum value of $p_0$ ($p_{0,opt}$) that results in the maximum turbine power output ($P_{max}$). In the cogeneration system for the optimum conventional evaporation process, $p_{0,opt}$ is 186.8 kPa, and $P_{max}$ is 29,286 kW. In the cogeneration system for the optimum modified evaporation process, $p_{0,opt}$ is 157.0 kPa, and $P_{max}$ is 29,442 kW. It is interesting to compare the cogeneration systems for the optimum modified evaporation process and a non-optimum conventional evaporation process, in which $p_0$ is 200 kPa. The non-optimum conventional process has the same juice processing capacity as the optimum conventional process, but it is less energy efficient. The value of $SE$ in this process is 2.411, and the value of $m_{v,0}$ is 43.31 kg/s. The turbine power output of the cogeneration system that uses this process is 28,789 kW, which is 2.3% lower than the turbine power output of the cogeneration system that uses the optimum modified

evaporation process. Table 1 shows simulation results of cogeneration systems for the non-optimum conventional evaporation process, the optimum conventional evaporation process, and the optimum modified evaporation process.

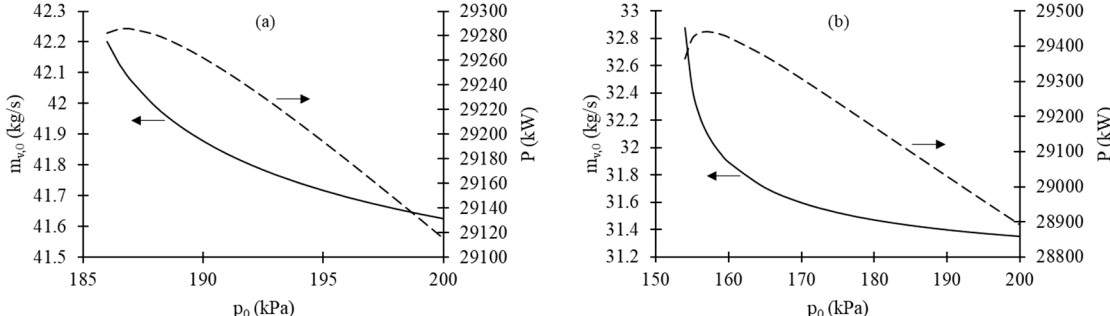

**Figure 6.** Variations with extracted steam pressure ($p_0$) of extracted steam consumption ($m_{v,0}$) and turbine power output ($P$) of the cogeneration systems that have the optimum distributions of evaporator surface areas for (**a**) the conventional evaporation process and (**b**) the modified evaporation process.

**Table 1.** Simulation results of cogeneration systems for the non-optimum conventional evaporation process, the optimum conventional evaporation process, and the optimum modified evaporation process.

| | Conventional EP | | Optimum Modified EP |
|---|---|---|---|
| | **Non-Optimum** | **Optimum** | |
| $A_1 (m^2)$ | 4695 | 6611 | 4634 |
| $A_2 (m^2)$ | 4266 | 1558 | 2409 |
| $A_3 (m^2)$ | 2729 | 1335 | 1932 |
| $A_4 (m^2)$ | 1310 | 3496 | 4025 |
| $A_{h,1} (m^2)$ | 80 | 469 | 1399 |
| $A_{h,2} (m^2)$ | 2420 | 2031 | 1101 |
| $p_0 (kPa)$ | 200.0 | 186.8 | 157.0 |
| $p_1 (kPa)$ | 150.0 | 150.0 | 122.6 |
| $p_2 (kPa)$ | 113.4 | 85.8 | 79.4 |
| $p_3 (kPa)$ | 81.5 | 44.2 | 44.6 |
| $p_4 (kPa)$ | 16.0 | 16.0 | 16.0 |
| $m_{f,in} (kg/s)$ | 125.0 | 125.0 | 125.0 |
| $m_{v,0} (kg/s)$ | 43.31 | 42.09 | 32.11 |
| $m_a (kg/s)$ | 13.16 [1] | 13.16 [1] | 13.16 [2] |
| $m_{fuel} (kg/s)$ | 21.00 | 21.00 | 21.00 |
| $P (kW)$ | 28,789 | 29,286 | 29,442 |

[1] Vapor bled from the first effect at 150 kPa. [2] Extracted steam from turbine at 150 kPa.

Table 1 shows that steam and vapor pressures in the optimum modified evaporation process are lower than those in the non-optimum and optimum conventional evaporation processes. Sucrose inversion losses in all effects of evaporators in the three processes are compared in Table 2. It can be seen that sugar inversion loss is largest in the first effect of each process. Sucrose inversion loss in the first effect of the optimum modified evaporation process has the lowest value due to the smallest extracted steam pressure and temperature. As a consequence, the total sucrose inversion loss of the optimum modified evaporation process is 63% lower than that of the non-optimum conventional evaporation process.

**Table 2.** Comparison of sucrose inversion losses in the non-optimum conventional evaporation process, the optimum conventional evaporation process, and the optimum modified evaporation process.

| Effect Number | Conventional EP | | Optimum Modified EP |
|:---:|:---:|:---:|:---:|
| | **Non-Optimum** | **Optimum** | |
| 1 | $2.95 \times 10^{-3}\%$ | $4.16 \times 10^{-3}\%$ | $1.58 \times 10^{-3}\%$ |
| 2 | $1.79 \times 10^{-3}\%$ | $2.93 \times 10^{-4}\%$ | $3.29 \times 10^{-4}\%$ |
| 3 | $6.38 \times 10^{-5}\%$ | $5.50 \times 10^{-5}\%$ | $7.46 \times 10^{-5}\%$ |
| 4 | $6.15 \times 10^{-6}\%$ | $1.59 \times 10^{-5}\%$ | $1.77 \times 10^{-5}\%$ |
| Total | $5.38 \times 10^{-3}\%$ | $4.52 \times 10^{-3}\%$ | $2.00 \times 10^{-3}\%$ |

## 6. Conclusions

The comparison between the cogeneration system that used the conventional evaporation process and the cogeneration system that used the modified evaporation process was investigated in this paper. Bled vapor and steam extracted from the turbine were used, respectively, by the first and the second systems for heating duty in pan stages. Both conventional and modified evaporation processes had the total evaporator surface area of 13,000 m$^2$ and total juice heater surface area of 2500 m$^2$. They were designed to process 125 kg/s of inlet sugar juice. The distribution of evaporator surface area of the optimum modified evaporation process resulted in the maximum steam economy. The pressures of extracted steam supplied to the optimum modified evaporation process were chosen so that the turbine power output of the cogeneration system that used this process was maximized. According to simulation results obtained from the mathematical models developed for this investigation, extracted steam at a mass flow rate of 31.53 kg/s and a pressure of 157.0 kPa was required for the evaporator of the optimum modified evaporation process, and extracted steam at a mass flow rate of 13.16 kg/s and a pressure of 150.0 kPa was required for the pan stage of this process. The turbine power output was 29,442 kW for the cogeneration system that used the optimum modified evaporation process. This power output was 2.3% larger than the power output of the cogeneration system that used a non-optimum conventional evaporation process. Furthermore, since the pressure profile in the evaporator of the optimum modified process was lower than that of the non-optimum conventional process, sucrose inversion loss in the modified process was 63% lower.

**Funding:** This research received no external funding.

**Conflicts of Interest:** The authors declare no conflict of interest.

## Nomenclature

| | |
|:---|:---|
| $A$ | heat transfer surface of evaporator, m$^2$ |
| $A_h$ | heat transfer surface of juice heater, m$^2$ |
| $c_p$ | specific heat capacity, kJ/kg·°C |
| $h$ | specific enthalpy, kJ/kg |
| $I$ | mass fraction of lost sugar due to inversion |
| $m$ | mass flow rate, kg/s |
| $P$ | turbine power output, kW |
| $p$ | pressure, kPa |
| $SE$ | steam economy |
| $T$ | temperature, °C |
| $t$ | retention time, min |
| $U$ | heat transfer coefficient, kW/m$^2$·°C |
| $x$ | concentration of sugar juice, % |

**Greek Symbols**

| | |
|:---|:---|
| $\varepsilon$ | heat loss coefficient in evaporator |
| $\eta_\tau$ | turbine efficiency |
| $\rho$ | density, kg/m$^3$ |

**Subscripts**

| | |
|---|---|
| *a* | vapor to pan stage |
| *b* | vapor to juice heater |
| *c* | vapor from flash tank, condenser |
| *e* | flash evaporation |
| *f* | sugar juice |
| *h* | juice heater |
| *i* | effect number |
| *l* | saturated liquid |
| *s* | steam |
| *v* | saturated vapor |
| *vl* | vapor-to-liquid |
| *x* | juice heating inside evaporator vessels |

**Superscripts**

| | |
|---|---|
| *in* | inlet of an effect |
| *out* | outlet of an effect |

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
