# Peer review of "Modification of Conventional Sugar Juice Evaporation Process for Increasing Energy Efficiency and Decreasing Sucrose Inversion Loss"

_processes, doi:10.3390/pr8070765_

Round 1

Reviewer 1 Report

Comments,
This manuscript describes the modification of the conventional sugar juice evaporation process for increasing energy efficiency and decreasing sucrose inversion loss. The authors investigated the effects of the evaporation process, and they modified modification this process, with a novel modified evaporation process yields 2.3% more power output than the cogeneration system that uses a non-optimum conventional cogeneration process. These results are practical and could be the interest of most of the readers in the sugar process field. Therefore, the present manuscript is suitable for the Process journal.

Author Response

The reviewer suggests moderate English changes. The revised manuscript has, therefore, been edited to improve its English language and style. 

Reviewer 2 Report

Dear Authors,

thank you a lot for the interesting paper. The subject is very up-to-date. In every industry scientists look for improvements to make processes more productive, less material-consuming, less capital-intensive, shorter etc. You chose very specific process of food industry, e.g. Sugar Juice Evaporation Process. You compare Conventional and Modified evaporation processes. The paper was interesting and made in professional way.

You presented very specific and very narrow area of its use. That is why I gave average rating in "interest to readers" because it can be used and read mainly by people connected to maritime sector. "Originality / Novelty", "Significance of Content" and "Scientific Soundness" I assessed very high. "Quality of Presentation" I assessed as average because I found some small mistakes. However, I have to emphasize that these are not substantive mistakes. My comments will help to improve your paper:

  1. You submitted paper to special issue: Industrial Chemistry Reactions: Kinetics, Mass Transfer and Industrial Reactor Design. Maybe add 1-2 keywords proposed by editors for this SI.
  2. At the end of Introduction add paragraph with paper's structure so readers know what to expect and which order.
  3. Equations: check numbering (all equations). In pdf file some brackets are too close to each other or almost on the symbols. Please check it. I am old I have some difficulties reading them (eq. 1-11). Maybe it is only a problem of pdf, I have no idea.
  4. Figure 3-6. You have some elements and you use "a)", "b)" ect... In Figure 3 you placed these symbols in different way than in other Figures. Please unify it.
  5. Any limitation of the research? Please add it in Conclusion.

Besides these comments I think your paper can be published by MDPI. Of course after correction introduced.

Author Response

  1. Two additional keywords (Process design and Mass transfer) have been added in the revised manuscript, and shown in yellow background.
  2. The last paragraph of Section 1 has been rewritten in accordance with the reviewer's suggestion. The change is highlighted with yellow background.
  3. This problem will be dealt with in the edited version.
  4. Designations (a) and (b) have been placed at the top of Fig. 3 like Figs. 4-6.
  5. I can't think of any limitation. In fact, the modification can be easily implemented in an existing process with minimal costs. I sincerely hope to see the implementation of this modification in the sugar industry in the near future.